# The Paleomineralogy of the Hadean Eon Revisited

**DOI:** 10.3390/life8040064

**Published:** 2018-12-17

**Authors:** Shaunna M. Morrison, Simone E. Runyon, Robert M. Hazen

**Affiliations:** 1Geophysical Laboratory, Carnegie Institution for Science, 5251 Broad Branch Road NW, Washington, DC 20015, USA; rhazen@carnegiescience.edu; 2Department of Geology and Geophysics, University of Wyoming, Laramie, WY 82071, USA; srunyon@uwyo.edu

**Keywords:** mineral evolution, photo-oxidation, impacts, paleomineralogy, Hadean Eon

## Abstract

A preliminary list of plausible near-surface minerals present during Earth’s Hadean Eon (>4.0 Ga) should be expanded to include: (1) phases that might have formed by precipitation of organic crystals prior to the rise of predation by cellular life; (2) minerals associated with large bolide impacts, especially through the generation of hydrothermal systems in circumferential fracture zones; and (3) local formation of minerals with relatively oxidized transition metals through abiological redox processes, such as photo-oxidation. Additional mineral diversity arises from the occurrence of some mineral species that form more than one ‘natural kind’, each with distinct chemical and morphological characteristics that arise by different paragenetic processes. Rare minerals, for example those containing essential B, Mo, or P, are not necessary for the origins of life. Rather, many common minerals incorporate those and other elements as trace and minor constituents. A rich variety of chemically reactive sites were thus available at the exposed surfaces of common Hadean rock-forming minerals.

## 1. Introduction

The paleomineralogy of Earth’s earliest half-billion years, though poorly preserved in the known rock record, is important for understanding key evolutionary episodes, including the rate of planetary differentiation, the initiation of plate tectonics, the rate and nature of continental crust formation, and the origins of life. Accordingly, Hazen [1] proposed a preliminary list of 420 plausible mineral species that were likely present during Earth’s Hadean Eon. These minerals are known to arise through purely physical and chemical processes—mechanisms such as fractional crystallization, regional metamorphism, serpentinization, evaporite crystallization, and hydrothermal alteration that are thought to have been active during Earth’s first 500 million years, prior to the origin and global distribution of life. 

Knowledge of the diversity and distribution of Hadean minerals is especially important for models of life’s origins. Numerous mineral species and mineral groups—including oxides, silicates, carbonates, phosphates, sulfides, borates, and molybdates—have been invoked in models of life’s origins [2,3,4,5,6,7,8,9,10,11,12,13,14,15,16,17]. The varied roles proposed for these minerals include catalysis for synthesis of essential biomolecules; protection of those molecules through surface interactions; molecular selection and concentration of specific subsets of molecules, including chiral selection, from a dilute prebiotic milieu; templating and formation of biopolymers; formation of lipid membranes; and surface redox chemistry that promoted life’s earliest metabolism (e.g., [18,19,20,21]). It has been assumed that mineral species proposed in these origins-of-life scenarios must have been present in the near-surface environment of prebiotic Earth for the scenario to be valid.

In this contribution, we first re-examine the list of plausible Hadean minerals proposed by Hazen [1], expanding the list as a consequence of three important insights. The first group of plausible prebiotic mineral species on Earth—some of which are not known in Earth’s near-surface environment today—include a variety of crystalline organic compounds, including amino acids, nucleobases, hydrocarbons, co-crystals, clathrates, and other species that could have occurred but have long since been consumed by ubiquitous cellular life [22,23].

Second, the profound potential influence of large bolide impacts on mineralization, though previously recognized as a possibility [24,25], has been elaborated by the work of Glikson and Pirajno (2018). In particular, we must consider additional mineral species formed in local hydrothermal systems related to impacts.

The third possible expansion of the list of Hadean minerals results from increasing recognition of abiological mechanisms for creating significant redox gradients at local, regional, and global scales. The prebiotic near-surface redox state of Earth was controlled primarily by the coexistence of both ferrous and ferric iron—i.e., conditions close to the hematite-magnetite oxygen buffer. However, significant excursions from that buffered state by several plausible natural processes may have led to the formation of minerals at much higher or lower oxygen fugacities.

Given these additional modes of mineral formation—parageneses not fully considered among the 420 species tabulated in the preliminary list of Hazen [1]—we catalog scores of additional species that might have occurred on Earth prior to life’s origins. In addition, we suggest that the nature of mineral diversity, itself, needs to be reassessed based on the emerging concept of “natural kind clustering” applied to mineralogy [26]. Such common mineral species as calcite, quartz, hydroxylapatite, and pyrite can form through varied physical, chemical, and biological mechanisms—processes that result in distinct chemical and morphological characteristics of individual samples. In the context of planetary evolution, each different mode of formation of a mineral species should be treated as a distinct ‘natural kind’, with its own contextual story to tell.

Finally, we note that the absence of certain rare minerals in the Hadean Eon, for example species incorporating essential boron, molybdenum, or phosphorus, need not be an impediment to origins-of-life scenarios that invoke exotic mineral catalysts (e.g., [15]). These and many other rare elements are ubiquitous as trace and minor elements in rock-forming minerals. Therefore, numerous types of chemically reactive sites have been exposed at the surfaces of Earth’s commonest minerals throughout geological history. 

## 2. Prebiotic Organic Materials

Carbon is a relatively minor constituent of Earth’s crust—less than 0.1 weight percent, most of which today is stored in extensive platforms of carbonate minerals [27,28,29]. However, recent speculations on the mineralogy of Saturn’s large, hydrocarbon-rich moon Titan [22], suggest that organic crystals can be diverse and abundant, especially in an environment not subjected to predation by cellular life [23].

Most of the more than 50 ‘organic minerals’ that have been identified on Earth today [30,31] are biologically derived—crystals formed by the subaerial alteration of excrement (notably in cave guano deposits), from decaying vegetation and animal matter, or derived from coal and oil shale (e.g., [30,32,33]), including crystals that form near coal fires from condensing hot vapors [34,35,36,37,38,39]. Benner et al. [40] emphasize that many of these compounds are highly oxidized oxalates and carboxylates, which have low value as nutrients in the modern oxygen-rich environment and thus may persist in the presence of cellular life. None of these extant organic phases were included in the preliminary list of Hadean minerals.

By contrast, many organic crystals that might have occurred on prebiotic Earth, either by endogenous synthesis or exogenous delivery, represent food that today would be consumed quickly. Early in Earth’s history, when carbon-rich chondrites and comets transported abundant organic molecules to the sterile surface [41,42,43,44,45,46] and subsurface organic synthesis occurred in carbon-rich aqueous fluids [47,48,49,50,51,52], numerous now extinct organic crystals may have existed. Table 1 lists representative potential organic compounds that might have occurred as crystalline minerals on prebiotic Earth. 

While details of Hadean organic materials will likely remain speculative, concentration of soluble molecular species in shallow water environments, followed by evaporation and precipitation, could have led to crystals of the most abundant amino acids, including glycine, alanine, and aspartate [46]; nucleobases such as adenine, guanine, and uracil [45]; carbohydrates [53,54]; varied hydrocarbons including alkanes, alkenes, and polycyclic aromatics; and other key biomolecules (Table 1). Of note is the possibility that some organic molecules precipitated as chirally pure ‘right-handed’ and ‘left-handed’ crystalline forms [55,56,57,58]. Furthermore, given the widespread influence of volcanic activity and associated vents, a variety of organic crystals, including polycyclic aromatic and aliphatic hydrocarbons, may have condensed from a hot gas phase as they do near coal fires today. 

Maynard-Caseley et al. [22] also explored the fascinating diversity of organic ‘co-crystals’ with two or more small molecular building blocks. Their proposed examples focused on phases such as 1:1 acetylene:methanol, 2:1 acetylene:acetonitrile, and 3:1 benzene:ethane, consistent with 92 K surface environment of Titan. The possibility of higher-temperature molecular co-crystals, especially disordered sub-crystalline arrangements of varied lipid or polycyclic aromatic hydrocarbon molecules, should be considered. Organic-enclosing water- or silica-based clathrate compounds at low temperatures [59,60,61], including clathrates intercalated with clay minerals [62], represent additional potential Hadean species. 

Note that prebiotic organic compounds have been invoked in chemical scenarios of life’s origins [40,63]. Benner et al. [40] consider reactive carbohydrates and related nitrogen-bearing compounds, while Ritson and Sutherland [64] focus on RNA assembly in the presence of crystalline ferrocyanide—phases that should be added to lists of plausible Hadean minerals. In addition, Sutherland and colleagues have studied 2-aminooxazole, a plausible prebiotic organic compound that can serve as an intermediate molecule in RNA synthesis [65,66].

An additional intriguing possibility is that a complex suite of organic microcrystals still exists in carbonaceous chondrite meteorites, which may have experienced extreme dehydration and thus incorporate a rich variety of organic and inorganic “salts” [67]. Standard analyses would not easily detect these submicroscopic crystalline zones, so the opportunity for new discoveries of preserved ancient organic crystals may exist.

## 3. Impact Mineralization

Major impacts of asteroids and comets have the potential to affect the mineral diversity of a terrestrial planet or moon in at least four ways. On the one hand, such impacts may have temporally reduced mineral diversity, at least at a local scale, by disrupting the crust and melting and/or vaporizing minerals near the impact site. Hadean Earth may have experienced tens of thousands of impacts of objects greater than 1 km in diameter, with hundreds of impacts greater than 10 kilometers in diameter [68]. Such events represented significant ‘reset buttons’ for local mineralogy.

On the other hand, large impacts likely enhanced mineral diversity in three significant ways. (1) The presence of high-pressure shock minerals, such as the ‘lonsdaleite’ allotrope of carbon and the dense coesite and stishovite forms of SiO_2_, are commonly associated marker minerals for impact events. (2) Large impacts have the potential to excavate and scatter rocks from deep terrains, thus bringing to the surface high-pressure-temperature metamorphic minerals that would otherwise not be seen. (3) Impacts can create significant hydrothermal systems in circumferential fracture zones that persist for upwards of 100,000 years, thus creating local mineralizing environments (e.g., [25]).

The first two of these mechanisms were recognized and incorporated by Hazen [1] in a list of plausible Hadean minerals. However, the possibility of extensive hydrothermal zones, though previously explored by several authors [25,69,70], was less fully developed. New contributions, summarized by Glikson and Pirajno (2018), elaborate on this idea and point to several likely Hadean minerals that should be added to the list (Table 2).

The significance of impact-produced fracture zones lies in their potential geographic and temporal extent. Deep circumferential hydrothermal systems up to hundreds of kilometers in diameter could have persisted for hundreds of thousands of years [25,69,70]. These large-scale systems of fluid–rock interactions likely pre-dated significant hydrothermal systems associated with subduction and arc volcanism. The mineralogical consequences of such Hadean impact-related processes remain speculative, but the occurrence of significant local hydrothermal alteration, including production of zeolites and clay minerals, seems likely [71,72]. In addition, the selection and concentration of relatively rare chemical elements by hydrothermal fluids would have led to the emplacement of ore bodies, including those featuring abundant chalcogenide and sulfosalt minerals. More recent examples of impact-associated ore deposits, though probably not exact analogies of the Hadean context, have been documented by the Ni-Cu mineralization of the Sudbury Igneous Complex in Canada [73], Pb-Zn-Ag-Ba ores of the Siljian Ring Complex in Sweden [74], and several Australian mines [68,72]. 

The question of what new minerals might have arisen as a consequence of these localized hydrothermal systems must remain speculative. However, a major impact on mafic or ultramafic lithologies, especially when coupled with the input of a bolide concentrated in iron, nickel, and other siderophile elements, as well as sulfur and other chalcophile elements, points to a potentially richer assembly of primary and secondary minerals than suggested by Hazen (2013). Notable additions to the preliminary list of 420 species could include minerals of first-row transition metals, including Ti, V, Cr, Mn, Fe, Co, Ni, Cu, and Zn, as well as As, Se, Mo, Ag, Sb, Au, Pb, Bi, and platinum group elements—elements commonly associated with sulfide deposits. Accordingly, Table 2 lists an additional 33 plausible Hadean mineral species, all of which are less common than those proposed by Hazen (2013), but which are frequently associated with hydrothermal ore bodies today.

Among the 33 additional possible Hadean mineral species in Table 2 are relatively common sulfides, chalcogenides, and sulfosalts—primary hydrothermal minerals that incorporate elements that might have been mobilized in large impact zones. An additional 25 oxysalt and halide minerals, most of them secondary, appear in Table 3. Of special interest in terms of plausible Hadean hydrothermal minerals are molybdates, which have been proposed by Benner and colleagues [16] to play special chemical roles in catalytic synthesis of carbonhydrate precursors of ribonucleosides and nucleotides. In the previous list, molybdenite (MoS_2_) was the only Hadean molybdenum-bearing mineral. Most molybdates are known principally as secondary minerals in the oxidized zones of ore bodies, and thus were excluded from consideration. However, here we consider the possibility of powellite (CaMoO_4_), a mineral known to occur in association with basalt, tachite, and granite pegmatite [75]. 

## 4. Prebiotic Redox Gradients on Earth

Planets create redox gradients at varied scales from local to global. An assumption underlying virtually all hypotheses of life’s origins and evolution is that ancient life exploited those gradients as a source of chemical potential energy. Only much later did life, especially cells engaged in oxygenic photosynthesis, begin to create and sustain significant redox gradients of their own. The hypothesis that thousands of oxidized mineral species represent a global-scale mineralogical biosignature (e.g., [76]) is based on the assumption that significant near-surface redox gradients did not exist before the ~2.5 Ga ‘Great Oxidation Event’. However, this assumption might be challenged on the grounds that planets can display purely physico-chemical processes that create redox gradients at various scales, thus potentially generating extremely oxidized or reduced assemblages of minerals as electrons are shifted from a more oxidized to a more reduced environment. 

Explorations of the nature and extent of prebiotic redox gradients on Hadean Earth deserve a detailed quantitative analysis with estimates of the magnitude and duration of such redox excursions. Here we present a qualitative summary and suggest a number of electron transfer mechanisms by which such redox gradients may have arisen before the origins of life. For convenience, such mechanisms can be divided into global-, regional-, and local-scale phenomena.

### 4.1. Global-Scale Redox Gradients 

At the largest-scale of planetary differentiation, the immiscibility of silicate- and metal-rich melts, and the consequent gravitational segregation of a metal-rich core, creates a significant redox gradient at the core-mantle boundary and a permanent oxidation of Earth’s outer layers relative to the core. This mechanism, by which electrons are sequestered in the native iron-nickel alloys of the deepest interior, suggest that Earth’s mantle approached its current oxidation state by 3.5 to 4.0 Ga. Geochemical evidence from both isotopes and trace elements (e.g., [77,78,79,80,81,82]), coupled with the pressure effect on oxygen fugacity (e.g., [83,84]), suggest that biology did not play a role in this planetary-scale interior redox distribution. 

Additional large-scale development of redox gradients can occur when upper atmosphere photo-dissociation of water and subsequent hydrogen escape to space leads to a gradual increase in atmospheric oxygen fugacity. The ~0.15 mole fraction of molecular oxygen in the martian atmosphere is a consequence of this planetary-scale process [85]—a physicochemical effect that creates a redox gradient independent of life [86]. Global changes in atmospheric redox might also occur through “impact erosion” [87]. Nevertheless, most models of Earth’s Hadean atmosphere suggest that it was significantly reduced compared to today’s atmosphere [88,89,90,91], with a surface environment sustained close to the hematite-magnetite buffer (i.e., *f*O_2_ ~ 10^−72^; e.g., [92,93]). These global-scale redox gradients were incorporated in the analysis of Hazen [1] and do not lead to any new predicted prebiotic mineral species.

### 4.2. Regional-Scale Redox Gradients

Regional-scale (>1 km) movements of fluids and crystals can generate redox gradients in a variety of ways [94,95,96]. For example, redox gradients arise from crystal fractionation and settling of denser Fe^2+^-rich olivine during the formation of dunite [97,98,99,100,101,102]. This process creates a redox gradient at the scale of a magma chamber. Similarly, the crystallization of magnetite within a magma body results in redox changes of a melt with the increased rate of total Fe and particularly Fe^3+^ removal from a magma (e.g., [103]). This redox change due to magnetite crystallization is commonly accompanied by abrupt sulfur saturation and can potentially lead to the formation of an immiscible sulfide melt (e.g., [104,105]).

Many examples of immiscibility within natural magmatic systems are known, and compositional differences resulting from unmixing can be extreme (e.g., [106]), potentially creating redox gradients. Immiscibility between felsic and mafic silicate liquids has been documented in layered mafic intrusions, anorthosite complexes, mid-ocean ridge magma chambers, and granitoids (e.g., [107]).

Even in situations in which one immiscible phase is volumetrically minor compared to the other, it may be extremely effective in extracting and transporting large amounts of trace and rare elements, such as the case for immiscible sulfide melts concentrating PGEs and other chalcophile elements such as Cu and Ni from a larger silicate melt body [106,108,109,110]. The more reduced sulfide melts, segregated from more oxidized silicate melts, are commonly called upon as an important ore-forming process in magmatic Ni-Cu ± PGE deposits [110,111,112,113,114,115,116].

Similarly, carbonate-silicate liquid immiscibility has been invoked in the formation of carbonatite magma (e.g., [117,118,119,120]), though the extent to which such immiscibility leads to redox gradients is not fully resolved. Na, Ca, P, and F partition into carbonate melt, while Mg, Mn, Fe, Al, Si, and Ti concentrate in the silicate melts. Some experimental work has shown that oxidized sulfur (SO_4_^2−^) and uranium (U^6+^) preferentially partition into carbonate melt (along with Ca, P, Nb, Ta, and Pb), whereas reduced sulfur and uranium partition into silicate melt [120]. However, other analyses point to the derivation of more reduced carbonate from more oxidized silicate melts [109,121,122,123,124,125,126]. 

Another controversy relates to the possibility that anorthosite and other mafic/ultramafic silicate melts may separate from immiscible Fe-Ti oxide melts [127,128,129,130,131,132]. Arsenide melt immiscibility is a special case that occurs when mafic or ultramafic magmas become saturated in As [133,134,135,136,137]. 

These regional processes not only create redox gradients as fluids and/or crystals separate, but they also have the potential to produce new mineral species in zones of relatively high or low oxidation states. On a potentially larger scale, recent experimental work has suggested that immiscibility between H_2_O and H_2_ may occur in the mantle and may provide a mechanism for the rapid oxidation of the upper mantle immediately following core formation (e.g., [138]). 

### 4.3. Local-Scale Redox Gradients

At least five non-biological, near-surface mechanisms, all prevalent during the Hadean Eon, would have produced significant redox excursions at the scale of millimeters to meters: (1) fine-scale redox gradients resulting from magma segregation; (2) meteorite falls; (3) lightning strikes, notably near active volcanoes; (4) seafloor venting of sulfide-rich hydrothermal fluids; and (5) photo-oxidation of exposed minerals. Meteorites, lightning, and hydrothermal vents would likely have created locally reduced environments, whereas photo-oxidation has the potential to promote excursions to significantly more oxidized mineral-forming environments.

#### 4.3.1. Local Immiscibility

Just as immiscibility (or magma unmixing) may form regional-scale redox gradients, separation of fluids may be a powerful mechanism for forming local redox gradients. Recent experimental studies (e.g., [139,140]) have created localized redox gradients at the scale of hundreds of microns to millimeters. For example, redox gradients are proposed to form due to ferrous iron liberation during clinopyroxene and orthopyroxene dissolution in a more mafic melt, coupled with the diffusion of that ferrous iron into a more felsic melt. Local redox gradients are enhanced by the charge imbalance resulting from diffusion of charged ions across a magma mixing interface [139]. These localized redox gradients may be large (e.g., almost two orders of magnitude fO_2_), develop rapidly, and remain in this dynamic equilibrium on a time scale of days to potentially weeks [140]. Felsic magmas were unlikely to be common in the Hadean Eon. However, the occurrence of ancient detrital zircon grains in the Jack Hills sediments of Western Australia supports the existence of granitic crust on earth as early as 4.4 Ga [141,142,143]. Whether these features formed from reworking through interaction with preexisting Hadean crust or from impact melting (e.g., [144,145]), the possibility exists for development of local-scale redox gradients forming from magma mixing events in the Hadean Eon.

#### 4.3.2. Meteorites

Relatively reducing environments would have occurred in the vicinity of falls of certain classes of meteorites, including irons, stony irons, and enstatite chondrites. Enstatite chondrites, in particular, incorporate a suite of extremely reduced mineral phases, including silicides, phosphides, nitrides, and carbides. Such phases in a shallow aqueous environment create environments of extreme chemical disequilibrium that are unlikely to be caused by other near-surface phenomena. Pasek and colleagues [13,146,147,148] have focused special attention on prebiotic aqueous reactions between various organic species and schreibersite [(Fe,Ni)_3_P], a common meteorite phosphide mineral.

#### 4.3.3. Lightning Strikes

Lightning strikes, though highly localized, are responsible for significant mineralogical novelty as violent pulses of electrons and associated heating to >2000 K can cause dielectric breakdown of rock-forming minerals. The associated conditions are extremely reducing, approximating the SiO_2_-Si buffer [149,150]. The most familiar mineralogical consequences of lightning are fulgurites, which form when cloud-to-ground lightning alters soils and rocks. Among the mineralogical novelties found in some fulgurites are such reduced phases as native silica and a variety of iron silicides [149,151], reduced phosphorus compounds [152,153], and fullerene-type allotropes of carbon [154]. Of special note in the context of prebiotic chemistry are the effects of volcanic lightning [155], which would have been more prevalent in the Hadean Eon.

#### 4.3.4. Hydrothermal Vents 

Ocean floor hydrothermal vents through mafic and ultramafic lithologies represent a net electron flow from mantle to crust, with measured potentials up to 600 millivolt at an active vent [156]. Furthermore, Ang et al. [157] observed direct thermoelectric conversion in natural chalcopyrite (CuFeS_2_) from a sulfide-rich ‘black smoker’ hydrothermal vent system in the southern Lau Basin at the Australian-Pacific plate boundary. The resulting sulfide-rich metal deposits incorporate a variety of transition metal minerals [158,159,160] and represent a significant worldwide economic resource [159,161].

#### 4.3.5. Photo-Oxidation

The examples above describe local-scale reducing conditions caused by immiscible fluids/magmas, meteorites, lightning, or hydrothermal vents. None of these processes is likely to result in significant additions to the lists of plausible Hadean minerals. By contrast, photo-oxidation by exposure to ultraviolet radiation at Earth’s unshielded ancient surface, though poorly studied in the context of mineral evolution, might have led to local production of more oxidized minerals—notably minerals that form at oxygen fugacities significantly greater than that of the hematite-magnetite buffer. 

In some instances, photo-oxidation of a mineral that incorporates a reduced transition metal can lead to a reduced product plus a new mineral with a more oxidized transition metal. For example, Kim et al. [162] report on the photo-oxidation of siderite to form hydrogen and ferric iron oxides—a reaction similar to that of UV irradiation of aqueous Fe^2+^ ions [163,164,165].

Special attention has been focused on possible photocatalytic roles of the surfaces of rutile and anatase polymorphs of TiO_2_ [166,167,168,169]. When exposed to UV radiation, these minerals have been shown to photo-oxidize water, as well as a variety of organic species. Titanium dioxide is also capable of oxidizing nitrogen compounds, including di-nitrogen gas, to produce nitrates and nitrites [170,171]—compounds that could lead to local concentrations of minerals such as niter [K(NO_3_)] and nitratine [Na(NO_3_)]. In this regard, the surface of Mars today provides a possible analog for early Earth, as photo-oxidation has evidently led to the production of peroxides and perchlorates in martian soils [172,173,174,175]. In spite of these studies, much work remains to document the mineralogical consequences of photo-oxidation.

### 4.4. ‘Natural Kinds’ of Hadean Minerals

All prior mineral evolution analyses, including discussions of the diversity of Earth’s Hadean mineralogy, were considered in the context of the formal definition of mineral species established by the International Mineralogical Association or “IMA” [176,177]. The IMA methodology assigns each mineral ‘species’ a unique combination of idealized end-member composition and crystal structure—a protocol that has led to the approval of more than 5400 mineral species (for a current list see: rruff.info/ima; [178]). 

As useful as this unambiguous classification system continues to be, it fails to adequately capture the true diversity of mineral ‘natural kinds’—the demonstrably distinct occurrences of minerals. In the context of planetary evolution, particularly when surveying the mineral diversity of a specific period of Earth history, at least three types of planetary materials are imperfectly catalogued by the present system of mineral classification [26]: (1) some distinct kinds of minerals have been lumped according to the IMA classification; (2) some natural kinds of minerals have been split into multiple species by the IMA classification; and (3) noncrystalline materials are not included. Consider examples of each of these situations.

#### 4.4.1. Minerals That Have Been Lumped Together

Some mineral species occur in multiple distinctive ‘natural kinds’. For example, the species ‘diamond’, which is defined by the IMA as pure carbon in the diamond crystal structure, forms in at least five distinct natural kinds [26,179,180]. The first mineral in the cosmos was nano-crystalline diamond that condensed from hot, carbon-rich plasma in the cooling atmospheric envelopes of energetic stars [76,181]. Those vapor-deposited diamonds still fall to Earth in presolar grains and as constituents of chondrite meteorites [182]. Those nanodiamonds differ from high-pressure diamonds that form in Earth’s mantle, including “Type I” diamonds in carbon-saturated aqueous fluids and “Type II” diamonds in iron-nickel melts [180,183]. These forms in turn differ from diamonds formed by impact processes, formerly known as “lonsdaleite” [184,185], and the enigmatic black, porous sintered diamond masses known as carbonado [186,187]. Thus, from a planetary evolution perspective the single mineral species ‘diamond’ is better treated as a suite of distinct natural kinds, each with its unique combination of morphology, trace elements, and other properties. Similar arguments can be made for numerous common minerals, including pyrite (FeS_2_), quartz (SiO_2_), calcite (CaCO_3_), and hydroxylapatite [Ca_5_(PO_4_)_3_OH]. 

This situation has relevance to the enumeration of Hadean minerals. We need to be concerned not simply with the number of IMA mineral species, but also the number of ‘natural kinds’ of minerals. In terms of life’s origins, different kinds of pyrite, quartz, calcite, hydroxylapatite, and clay may have played very different chemical roles. The enumeration of mineral natural kinds is in its infancy, but these efforts will inform and modify future descriptions of Hadean mineralogy.

#### 4.4.2. Mineral Kinds That Have Been Split

The convention that official mineral species must correspond to endmember chemical compositions has led to splitting of some natural kinds into multiple categories. A dramatic example is provided by the tourmaline group of minerals, represented by the general formula *XY*_3_*Z*_6_(*T*_6_O_18_)*V*_3_*W*, where sites *X*, *Y*, *Z*, *T*, *V*, and *W* each can hold multiple different elements ([188,189]; rruff.info/ima). As a result of natural compositional variations, this classification leads to more than 30 IMA approved species in the tourmaline group, with several other potential species pending. Given chemical zonation in individual mineral grains, it is not uncommon for a single crystal to hold two or more different tourmaline species (e.g., [188,190]), even if all tourmaline grains were formed in the same petrogenetic event. On the other hand, it is also possible for individual mineral grains to represent two different natural kinds—for example, a tourmaline grain with an igneous core and hydrothermally-deposited rim. A division of tourmaline into natural kinds might thus provide a more parsimonious description of the supergroup and would facilitate a more accurate understanding of boron mineral diversity and distribution [189]. 

Similar arguments can be made for many mineral groups, including the apatite group [191,192], the garnet supergroup [193], the amphibole group [194], the mica group [195], the oxide spinel group [196,197], and pyrite [198] and other iron sulfides, all of which were likely present in the Hadean Eon.

#### 4.4.3. Non-Crystalline Materials

The IMA convention on mineral classification considers only crystalline materials. This protocol precludes significant volumes of condensed matter in near-surface environments of Earth, Mars, and other terrestrial worlds. Thus, for example, volcanic glass (e.g., ‘obsidian’), significant components of coal, composite materials, organic mixtures, and common constituents of soil are not easily classified under present mineral classification protocols [199].

This situation is especially significant in the context of Mars. Analyses of soils by NASA’s Mars Science Laboratory reveal more than 50 weight percent amorphous materials in some soil samples [200]. If Mars today mimics aspects of Earth’s Hadean mineralogy (e.g., [61]), we may be ignoring a significant component of early Earth’s near-surface environment. We conclude that expansion of mineral classification to include materials lacking a strictly periodic atomic structure will facilitate the characterization of the actual materials that make up planets.

## 5. Conclusions: Minerals and Origins-of-Life Chemistry

An important motivation for investigating the diversity and distribution of Hadean minerals is the presumed role of specific minerals in life’s geochemical origins. Recent experiments, notably by Steven Benner and colleagues, have documented significant plausible prebiotic catalytic roles for minerals of relatively rare elements, such as boron, molybdenum, and phosphorus [11,15,16]. However, it is not obvious that such minerals as powellite, colemanite, or lüneburgite would have formed during Earth’s first 500 million years. Even if these types of minerals were present at a few widely dispersed localities, the total volumes of these phases would have been trivial in the global context. As a consequence, a number of elegant chemical solutions to key steps in life’s origins have been challenged by the implausibility of invoking rare mineral species (e.g., [1,201]).

A solution to this impasse is to be found in the ubiquity of trace and minor elements in common rock-forming minerals. For example, the average crustal abundance of boron is ~10 ppm [202], with concentrations in the feldspar group of aluminosilicates reported as high as 5000 ppm as boron substitutes for aluminum [203]. Feldspar group minerals are the most abundant phases in Earth’s crust, accounting for as much as 60% of the volume of near-surface rocks [202]. As a consequence, immense numbers of borate-like surface sites are available to promote chemical reactions, even if no borate minerals *per se* are present.

Similar arguments can be presented for potentially reactive molybdate, phosphate, or any other surface sites, including sites where two or more atoms are juxtaposed in specific geometries. Indeed, for most relatively rare chemical elements the vast majority of atoms are to be found as trace or minor elements in common minerals, rather than in minerals with those elements as essential constituents. For example, Hazen et al. [204] calculate that all known reserves of cobalt minerals in major ore deposits represent less than one part in 50,000,000 of the cobalt atoms present at ~50 ppm in the vast expanses of basalt that pave the ocean floor [205]. 

We conclude that, in spite of the relative mineralogical parsimony of Earth’s first 500 million years, virtually any type of chemically reactive surface site, containing any of the more than 70 essential mineral-forming elements, would have been widely available throughout the Hadean Eon.

## Figures and Tables

**Table 1 life-08-00064-t001:** Examples of representative crystalline organic compounds that might have occurred during Earth’s Hadean Eon (>4.03 Ga) as a consequence of prebiotic endogenous production or exogenous delivery of soluble or volatile organic molecules, followed by molecular selection, concentration, and crystallization.

Molecular Class	Name
Amino acids	glycine
	alanine (D and L)
	β-alanine
	isovaline
	aspartate (D and L)
Nucleobases	adenine
	guanine ^1,^*
	uracil
Polycyclic Aromatic Hydrocarbons	fluorene ^2,^*
	anthracene ^3,^*
	pentacene ^4,^*
	coronene ^5,^*
	pyrene
	chrysene
	tetracene
Other Organics	ferrocyanide
	2-aminooxazole

* Compounds also known as organic minerals. ^1^ Guanine; ^2^ kratochvílite; ^3^ ravatite; ^4^ idrialite; ^5^ carpathite.

**Table 2 life-08-00064-t002:** Plausible Hadean Eon (>4.03 Ga) minerals: primary sulfides, chalcogenides, and sulfosalts that might be associated with hydrothermal fracture zones near large bolide impacts. The following minerals were not included in the preliminary list of Hazen (2013).

Group	Species	IMA Formula
Sulfides, Chalcogenides	argentopentlandite	Ag(Fe,Ni)_8_S_8_
	argentopyrite	AgFe_2_S_3_
	bismuthinite	Bi_2_S_3_
	breithauptite	NiSb
	clausthalite	PbSe
	dyscrasite	Ag_3+x_Sb_1-x_ (x ≈ 0.2)
	geffroyite	(Cu,Fe,Ag)_9_Se_8_
	maucherite	Ni_11_As_8_
	naumannite	Ag_2_Se
	siegenite	CoNi_2_S_4_
	stibnite	Sb_2_S_3_
Sulfosalts	aikinite	CuPbBiS_3_
	andorite	AgPbSb_3_S_6_
	berthierite	FeSb_2_S_4_
	boulangerite	Pb_5_Sb_4_S_11_
	bournonite	CuPbSbS_3_
	chalcostibite	CuSbS_2_
	cosalite	Pb_2_Bi_2_S_5_
	emplectite	CuBiS_2_
	jamesonite	Pb_4_FeSb_6_S_14_
	lindströmite	Pb_3_Cu_3_Bi_7_S_15_
	matildite	AgBiS_2_
	proustite	Ag_3_AsS_3_
	pyrargyrite	Ag_3_SbS_3_
	ramdohrite	Ag_3_Pb_6_Sb_11_S_24_
	samsonite	Ag_4_MnSb_2_S_6_
	seligmannite	CuPbAsS_3_
	smithite	AgAsS_2_
	sulvanite	Cu_3_VS_4_
	tealite	PbSnS_2_
	tetrahedrite	Cu_6_[Cu_4_(Fe,Zn)_2_]Sb_4_S_13_
	ullmanite	NiSbS
	zinkenite	Pb_9_Sb_22_S_42_

**Table 3 life-08-00064-t003:** Plausible Hadean Eon (>4.03 Ga) minerals: mostly secondary minerals, including those formed as a result of significant local redox excursions, for example those associated with photo-oxidation. The following minerals were not included in the preliminary list of Hazen (2013).

Group	Species	IMA Formula
Oxides, Hydroxides, Halides	akaganeite	Fe^3+^O(OH,Cl)
	arsenolite	As_2_O_3_
	cotunnite	PbCl_2_
	cuprite	Cu_2_O
	hibbingite	Fe^2+^_2_(OH)_3_Cl
	lawrencite	FeCl_2_
	zirkelite	(Ti,Ca,Zr)O_2-x_
Silicates	allanite-(La)	CaLa(Al_2_Fe^2+^)[Si_2_O_7_][SiO_4_]O(OH)
	allanite-(Y)	CaY(Al_2_Fe^2+^)[Si_2_O_7_][SiO_4_]O(OH)
Carbonates	azurite	Cu_3_(CO_3_)_2_(OH)_2_
	malachite	Cu_2_(CO_3_)(OH)_2_
	pyroaurite	Mg_6_Fe^3+^_2_(CO_3_)(OH)_16_·4H_2_O
Sulphates, Arsenates, etc.	brochanthite	Cu_4_(SO_4_)(OH)_6_
	chalcanthite	Cu(SO_4_)·5H_2_O
	copiaipite	Fe^2+^Fe^3+^_4_(SO_4_)_6_(OH)_2_·20H_2_O
	erythrite	Co_3_(AsO_4_)_2_·8H_2_O
	jarosite	KFe^3+^_3_(SO_4_)_2_(OH)_6_
	magnesiocopiapite	MgFe^3+^_4_(SO_4_)_6_(OH)_2_·20H_2_O
	morenosite	Ni(SO_4_)·7H_2_O
	nickelhexahydrite	Ni(SO_4_)·6H_2_O
	römerite	Fe^2+^Fe^3+^_2_(SO_4_)_4_·14H_2_O
	scorodite	Fe^3+^(AsO_4_)·2H_2_O
Molybdate, Tungstate, Phosphate	powellite	CaMoO_4_
	scheelite	CaWO_4_
	xenotime-(Y)	YPO_4_

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
