# Peer review of "The Paleomineralogy of the Hadean Eon Revisited"

_life, 2018, doi:10.3390/life8040064_

Reviewer 1 Report

Comments on “The Paleomineralogy of the Hadean Eon Revisited”

The authors expand the list of likely (or at least plausible) species that might have been present near the surface of the early earth. Three means of production of new structures are considered: organic crystals may have existed before consumption by living systems; collisions may have transformed species; and photo-oxidation may have produced new species. The authors also encourage the classification of species according to their mode of formation, which may complement the classification by chemical composition.

The existence of organic crystals (now extinct owing to their food value) must be speculative, but the speculation can be based on simulations excluding living things, or natural processes occurring in sterile settings.

The impact of extraterrestrial objects can generate or excavate species stable at high pressure, and alter the environment so to enable unusual chemical processes in “fracture zones.” This review adds emphasis to the possibility of formation of hydrothermal structures and subsequent reactions.

Evolution (and life itself) is driven and made possible by energy gradients. Gradients call for heterogeneity; the first example with no reference to biology is the distinction in solubility between the metal-rich core and silicate-rich surface of the earth. Global energy gradients attending photoprocesses are also well-recognized. Regional analogs are presented in this review. Local crystal migration (density-driven settling and solubility-driven fractionation) is featured, along with side-effects of energetic events (lightning, local hotspots/volcanism, meteorite impacts).

The review provides a guide to scholarly work related to these incompletely explored pathways. Consistent with the organization of the review by means of formation, the authors encourage a more nuanced classification, wherein (for example) “diamond” can be characterized by several “natural kinds” distinguished by the means by which the structure is accomplished. Describing “natural kinds” can lead to perception of relations among species which are chemically or structurally distinct. “Natural kinds” can include non-crystalline (amorphous) materials.

One may fairly describe this report as a collation of established results rather than any contribution of new data; but the purpose is of course to organize established data in a way that suggests new questions and encourages new perspectives. This purpose is well served by this clearly presented report, which can encourage new and meaningful experiments and data. I recommend publication in LIFE.

Author Response

Thank you for your review!

Reviewer 2 Report

The paper “The paleomineralogy of the Hadean Eon revisited” suggest that three ‘processes’ that could influence the presence of near-surface minerals on the prebiotic earth, which have largely been ignored, should be considered when compiling lists of these minerals. These processes include phases that promote the formation of organic crystals, minerals formed as a consequence of bolide impacts, and formation of redox gradients. The article is well written, and the ideas are compelling and will be of interest to both geochemists and prebiotic chemists. I suggest acceptance of this manuscript after minor revision.

Comments:

1)    I am concerned with the use of the phrase “organic mineral” for many of the organic compounds that are described in this way. I consider organic minerals to be formed from ionic bonds, salts, such as calcium carbonate, not organic crystals except for some allotropes like diamond (I also do not think that “organic mineral” is a very useful term in general, organic salts or crystals—when appropriate—is better). I suggest that the writers clearly define what they mean by Organic Mineral, or just leave the term out altogether. The main point of this section appears to be how processes that are usually considered to be important for the formation of inorganic mineral can also lead to the isolation/sequestration of organic compounds (this will be clearer to chemists than trying to first convince them that uracil is a mineral). Two concerns:

i)               2-aminooxozoline is described as an organic mineral in the main text, but it is not one. It is an organic compound that will crystalize from solutions (or from sublimation) that contain mostly 2-aminooxozoline (not complex organic mixtures that would be expected in the origin of life). And while this compound is of particular interest to John Sutherland (because he needs this for his ‘prebiotic’ reaction of cytosine/uridine to work) it is not of too great significance to the field of prebiotic chemistry, as opposed to cyanates which are required for many reaction schemes that form molecules now present in biology.

ii)              Table 1 has a number of issues. First, the compounds with * are not minerals, they are compounds that can crystalize. Second, the list of organic compounds chosen is very poor. There are dozens and dozens of amino acids that have been reported to form in model prebiotic reactions and in meteorites (see work by Jeff Bada and Jim Cleaves or Orig. Life Evol. Biospheres v38.6 (2008) p469-488 for a current list of amino acids that could have been present on the prebiotic earth), and all of the nucleobases and many nucleobase analogs have also (see Figure 6 of Hud’s paper Isr. J. Chem. 5.8 (2015) p891-905 for complete list).

2)    I really enjoyed the section on Prebiotic Redox Gradients on Earth. The ideas of this section would be of great interest to a number of prebiotic chemists as many proposed model prebiotic reactions require sources for oxidation (for example many of the pathways in Sutherland’s 2015 Nature Chemistry paper). I do, however, fear that chemists will not see this very important work. This could be changed if you include proposed reactions by Sutherland, Matt Powner, Ram Krishnamurthy, Eschenmoser and others that require oxidation steps. I do understand that such a discussion may be outside the scope of this work – but it would be of significant value so keep it in mind in the future.

3)    The conclusion that mineral classifications should include those of complex aperiodic nature in addition to ones with strict periodic atomic structure is an excellent point. The same applies to all organic compounds that were described, it is unlikely that any amino acids or nucleobases would crystalize as pure compounds and would instead more likely be found in complex mixture of similar compounds. I suggest that the writers make note that what was describe for inorganic minerals should also be extended to organic mixtures and thus tie the first and last sections together.

Author Response

Thank you for your careful and insightful review. We have implemented your suggestions and the explanation thereof is below in italics. 

Reviewer 2 comments and author responses:
1) I am concerned with the use of the phrase “organic mineral” for many of the organic compounds that are described in this way. I consider organic minerals to be formed from ionic bonds, salts, such as calcium carbonate, not organic crystals except for some allotropes like diamond (I also do not think that “organic mineral” is a very useful term in general, organic salts or crystals—when appropriate—is better). I suggest that the writers clearly define what they mean by Organic Mineral, or just leave the term out altogether. The main point of this section appears to be how processes that are usually considered to be important for the formation of inorganic mineral can also lead to the isolation/sequestration of organic compounds (this will be clearer to chemists than trying to first convince them that uracil is a mineral). Two concerns:

i) 2-aminooxozoline is described as an organic mineral in the main text, but it is not one. It is an organic compound that will crystalize from solutions (or from sublimation) that contain mostly 2-aminooxozoline (not complex organic mixtures that would be expected in the origin of life). And while this compound is of particular interest to John Sutherland (because he needs this for his ‘prebiotic’ reaction of cytosine/uridine to work) it is not of too great significance to the field of prebiotic chemistry, as opposed to cyanates which are required for many reaction schemes that form molecules now present in biology.

ii) Table 1 has a number of issues. First, the compounds with * are not minerals, they are compounds that can crystalize. Second, the list of organic compounds chosen is very poor. There are dozens and dozens of amino acids that have been reported to form in model prebiotic reactions and in meteorites (see work by Jeff Bada and Jim Cleaves or Orig. Life Evol. Biospheres v38.6 (2008) p469-488 for a current list of amino acids that could have been present on the prebiotic earth), and all of the nucleobases and many nucleobase analogs have also (see Figure 6 of Hud’s paper Isr. J. Chem. 5.8 (2015) p891-905 for complete list).

Replaced “organic minerals” with “organic materials,” “organic crystals,” “organic phases,” or “organic compounds,” except on line 83 when ‘“organic minerals”’ is used in reference to publications that specifically use this term. (Note: marking uracil as an organic mineral was a typo, we meant to mark guanine.)

Changed the caption of Table 1 to read “Examples of representative crystalline…” This list is intended to show only selected compounds that are known to crystallize and occur in relatively significant abundances.

2)  I really enjoyed the section on Prebiotic Redox Gradients on Earth. The ideas of this section would be of great interest to a number of prebiotic chemists as many proposed model prebiotic reactions require sources for oxidation (for example many of the pathways in Sutherland’s 2015 Nature Chemistry paper). I do, however, fear that chemists will not see this very important work. This could be changed if you include proposed reactions by Sutherland, Matt Powner, Ram Krishnamurthy, Eschenmoser and others that require oxidation steps. I do understand that such a discussion may be outside the scope of this work – but it would be of significant value so keep it in mind in the future.

This is a great suggestion - however, we agree with the reviewer that a more detailed discussion would be needed to include this information in the current manuscript and therefore we will keep it in mind for future, more-detailed manuscripts on organic mineralogy.

3) The conclusion that mineral classifications should include those of complex aperiodic nature in addition to ones with strict periodic atomic structure is an excellent point. The same applies to all organic compounds that were described, it is unlikely that any amino acids or nucleobases would crystalize as pure compounds and would instead more likely be found in complex mixture of similar compounds. I suggest that the writers make note that what was describe for inorganic minerals should also be extended to organic mixtures and thus tie the first and last sections together.

Non-crystalline section updated to read:

"Thus, for example, volcanic glass (e.g., “obsidian”), significant components of coal, composite materials, organic mixtures, and common constituents of soil are not easily classified under present mineral classification protocols."

Reviewer 3 Report

In the present manuscript, Morrison et al. revisit the previous list of plausible near-surface minerals present during the Earth’s Hadean Eon and suggest expanding it by including:

1)      prebiotic organic minerals. Inspired by the recently uncovered Titan’s near-surface mineralogy, the authors propose a list of prebiotic organic minerals that could have occurred in crystalline form such as amino acids, nucleobases, and aromatic hydrocarbons. The authors argue that these minerals are now extinct because they were quickly consumed when life originated on Earth. Occurrence of these mineral classes on early Earth are of particular interest to the origins of life research and the authors touch upon the subject by mentioning and including organic minerals that were involved in the prebiotic synthesis of RNA.

2)      primary hydrothermal minerals. Specifically, those that would have occurred in impact-related hydrothermal zones during the Late Heavy Bombardment period. Among the 33 newly identified plausible Hadean minerals is powellite, a novel primary mineral that may have played catalytic role in the prebiotic chemistry and thus essential to the origins of life.

3)      and additional 25 minerals which would result from photo-oxidation in local redox gradients.

Furthermore, Morrison et al. make a legitimate suggestion to overhaul/expand IMA mineral classification, in order to accommodate non-crystalline minerals and to distinguish between minerals with same chemical composition but with different occurrence and potential catalytic roles.

Finally, the authors argue that even if the mineral occurrence of low-abundance elements is unlikely in global scales, any type of reactive surface could virtually be widespread during the Hadean Eon.

In general, the manuscript is well-structured and very well-written. It is also extensive and balanced in its literature citations. I have only 2 minor comments:

-         Line 118. From an origins of life context, Hadean minerals are not only important for RNA synthesis, but also for membrane formation (Hanczyc, M. M., et al. 2003, Science, http://dx.doi.org/10.1126/science.1089904, Sahai N., et al. 2017, Scientific Reports, http://doi:10.1038/srep43418). Although Hanczyc et al. was cited in Hazen (2013), the second reference is new and shows that earliest oceanic and continental rocks (komatiite and tonalite, respectively) can also promote lipid membrane self-assembly ––in line with the authors narrative in the present manuscript.

-         Line 319. Another reference I find missing from this manuscript is Smirnov and Schoonen 2016, Elements, http://doi:10.2113/gselements.12.6.379). This is again a relatively new reference that is in line with the authors’ narrative where Smirnov and Schoonen showed that weathering of olivine produces awaruite (a mineral already present in the original 420 list).

Other than that, the manuscript was informative and a pleasure to read.

Author Response

Thank you for your careful review and recommendations. We have implemented your suggestions. Below, you will find our responses in italics.

Reviewer 3 comments:

-  Line 118. From an origins of life context, Hadean minerals are not only important for RNA synthesis, but also for membrane formation (Hanczyc, M. M., et al. 2003, Science, http://dx.doi.org/10.1126/science.1089904, Sahai N., et al. 2017, Scientific Reports, http://doi:10.1038/srep43418). Although Hanczyc et al. was cited in Hazen (2013), the second reference is new and shows that earliest oceanic and continental rocks (komatiite and tonalite, respectively) can also promote lipid membrane self-assembly ––in line with the authors narrative in the present manuscript.

This is a good example of minerals and the origin of life. I added it (with citations) to line 41-42.

-   Line 319. Another reference I find missing from this manuscript is Smirnov and Schoonen 2016, Elements, http://doi:10.2113/gselements.12.6.379). This is again a relatively new reference that is in line with the authors’ narrative where Smirnov and Schoonen showed that weathering of olivine produces awaruite (a mineral already present in the original 420 list).

Other than that, the manuscript was informative and a pleasure to read.

I added this reference to line 38.